# Influence of Lenvatinib on the Functional Reprogramming of Peripheral Myeloid Cells in the Context of Non-Medullary Thyroid Carcinoma

**DOI:** 10.3390/pharmaceutics15020412

**Published:** 2023-01-26

**Authors:** Chunying Peng, Katrin Rabold, Mihai G. Netea, Martin Jaeger, Romana T. Netea-Maier

**Affiliations:** 1Department of Internal Medicine, Division of Endocrinology, Radboud University Medical Center, 6525 GA Nijmegen, The Netherlands; 2Department of Internal Medicine and Radboud Center for Infectious Diseases, Radboud University Medical Center, 6525 GA Nijmegen, The Netherlands; 3Department of Immunology and Metabolism, Life and Medical Sciences Institute, University of Bonn, 53115 Bonn, Germany

**Keywords:** tyrosine kinase inhibitors, innate immunity, thyroid cancer, tumor-associated macrophages, cytokines

## Abstract

Lenvatinib is a multitarget tyrosine kinase inhibitor (TKI) approved for the treatment of several types of cancers, including metastatic differentiated thyroid cancer (DTC). The intended targets include VEGFR 1–3, FGFR 1–4, PDGFRα, RET, and KIT signaling pathways, but drug resistance inevitably develops and a complete cure is very rare. Recent data has revealed that most of the TKIs have additional ‘off-target’ immunological effects, which might contribute to a protective antitumor immune response; however, human cellular data are lacking regarding Lenvatinib-mediated immunomodulation in DTC. Here, we investigated in ex vivo models the impact of Lenvatinib on the function of immune cells in healthy volunteers. We found that monocytes and macrophages were particularly susceptible to Lenvatinib, while neutrophiles and lymphocytes were less affected. In tumor-immune cell co-culture experiments, Lenvatinib exerted a broad inhibitory effect on the proinflammatory response in TC-induced macrophages. Interestingly, Lenvatinib-treated cells had decreased cellular M2 membrane markers, whereas they secreted a significantly higher level of the anti-inflammatory cytokine IL-10 upon LPS stimulation. In addition, prolonged exposure to Lenvatinib impaired macrophages survival and phenotypical differentiation, which was accompanied by remarkable morphological changes and suppressed cellular metabolic activity. These effects were mediated by myeloid cell-intrinsic mechanisms which are independent of Lenvatinib’s on-target activity. Finally, using specific inhibitors, we argue that dual effects on p38 MAPK and Syk pathways are likely the underlying mechanism of the off-target immunological effects we observed in this study. Collectively, our data show the immunomodulatory properties of Lenvatinib on human monocytes. These insights could be harnessed for the future design of novel treatment strategies involving a combination of Lenvatinib with other immunotherapeutic agents.

## 1. Introduction

Differentiated thyroid cancer (DTC) constitutes the majority of thyroid carcinomas (TC) worldwide. Generally, treatment with surgery and radioactive iodine are effective for most cases. However, up to 15% of patients develop radioiodine-refractory metastatic disease (RAI-refractory-DTC) and have poor long-term prognosis [1]. Currently, the only therapeutic option available for these patients is systemic molecular targeted therapy, which is represented by tyrosine kinase inhibitors (TKIs). Lenvatinib, a multitargeted TKI against VEGFR 1–3, FGFR 1–4, PDGFRα, RET, and KIT signaling networks, has been approved for the treatment of metastatic DTC since 2015. Despite positive results from a SELECT trial, in a real-world setting, response to treatment differs among patients [2,3]. Oncological drug development is a time- and cost-consuming task, yet a high frequency of secondary mutations and paradoxical activation of alternative pathways often leads to drug resistance [3,4,5]. Therefore, innovative strategies to fully assess the antitumor activity of Lenvatinib are urgently needed.

Beyond targeting oncogenesis and angiogenesis processes, tumor immunotherapy remains a highly attractive yet not fully operational approach to treatment. Recent advances in immune checkpoint inhibitors (ICIs) offer a promising approach to achieve durable improvement by remobilizing the immune system against tumor proliferation and metastasis. However, the efficacy of ICIs is largely dependent on an immunogenic tumor microenvironment (TME), and, to date, ICI monotherapy has been proved effective only in a minority of solid cancers, as it largely blunted by the immunosuppressive factors with TME [6,7,8]. Myeloid cells constitute a highly plastic and heterogenous cell population in the tumor milieu and are often hijacked and reprogrammed by malignant cells to facilitate disease progression [9]. Notably, tumor-associated macrophages (TAMs) identified in advanced DTC display an immunosuppressive and tumor-promoting phenotype, and increased TAMs infiltration is correlated with poor prognosis [10,11,12]. This, in turn, makes them suitable targets for therapeutic intervention [13]. Interestingly, there has been emerging evidence indicating that most TKI therapies exert additional immunological effects via tumor intrinsic or extrinsic mechanisms [14,15,16,17,18]. Part of the immunomodulatory properties could be explained by their anti-angiogenesis effects via direct binding to the primary targets VEGFRs, which is also known as on-target activity. It should be noted that many FDA-approved TKIs interact with undesignated kinases/proteins expressed by healthy cells, termed as off-target effects. Interestingly, these off-target effects could lead to beneficial drug repurposing if the underlying mechanism could be characterized [19].

Recent results from different murine models revealed that an intact immune system was an essential prerequisite for the Lenvatinib-mediated tumor killing activity, and the combination with anti-PD-1 therapy significantly improved animal survival [20,21,22]. This is indicative of possible Lenvatinib-mediated immunomodulation, mainly involving the adaptive immune system. However, thus far, little has been known about the role of myeloid cells in these effects and how such a mechanism can be leveraged to improve clinical outcomes. Therefore, in this study, we set out to study whether Lenvatinib modulates the immune system in the context of TC and the potential mechanisms mediating these effects. This knowledge could contribute to the development of combination treatment strategies to enhance its effectivity and reduce toxicity. Our data show that ex vivo Lenvatinib treatment induced functional and metabolic reprogramming in circulating monocytes obtained from healthy volunteers, and this was mediated via dual inhibition of the p38 MAPK and Syk pathways.

## 2. Materials and Methods

### 2.1. Monocyte Isolation

Peripheral blood mononuclear cells (PBMCs) were isolated by density gradient centrifugation with Ficoll-Paque (GE healthcare, Diegem, Belgium) from buffy coats obtained from the Sanquin bloodbank, Nijmegen, The Netherlands. Percoll-monocyte enrichment was performed as previously described [23]. Briefly, PBMCs were layered on top of a hyper-osmotic Percoll solution, and the interphase layer was collected after centrifugation. Cells were resuspended in RPMI culture medium (Life Technologies, Carlsbad, CA, USA) supplemented with pyruvate (1 mM), glutamine (2 mM), gentamicin (50 µg/mL), HEPES (10 mM), glucose (5 mM), and 10% human pooled serum (HPS). In selected experiments, highly purified monocytes were obtained from PBMCs through negative selection using a human pan monocyte isolation kit, following the manufacturer’s instruction (Miltenyi Biotec, Bergisch Gladbach, Germany).

### 2.2. Neutrophil Isolation and ROS Production Analysis

Neutrophil isolation and functional analysis were performed as previously described [24]. EDTA-anticoagulated peripheral blood was collected from healthy donors with informed consent. After density-gradient centrifugation, red blood cells were lysed with hypotonic buffer, and the purity of the remaining neutrophils was measured on a Sysmex XN-450 Hematology Analyzer (Sysmex Corporation, Kobe, Japan). Reactive oxygen species (ROS) assays were performed with freshly isolated neutrophils using a luminescence assay in quadruplicates. A total of 200,000 neutrophils per well were added into a 96-well white assay plate (Corning, Corning, NY, USA). Due to the relatively brief lifespan of neutrophils, ROS production was measured after 1 h preincubation of Lenvatinib. Cells were stimulated with 0.5 μg/mL of phorbol myristate acetate (PMA) or plasma-opsonized Zymosan A (Sigma-Aldrich, St. Louis, MO, USA) at a concentration of 1 mg/mL. Luminol (Sigma-Aldrich) was added with a final concentration of 0.1 mM, and chemiluminescence was measured at a wavelength of 425 nm every 142 s for 1 h.

### 2.3. Cell Culture, Viability Test, and Stimulation

Several Lenvatinib concentrations within the clinically achievable dose range (100 nM to 1 µM) were tested in our experimental setup with both PBMCs and monocytes. Cell viability was determined by flowcytometry using fixable viability dye (FVD) eFlour 780 (Invitrogen, Carlsbad, CA, USA). After 24 h of incubation with Lenvatinib, the cells were labelled with FVD for 30 min and stained with surface marker. CD14^+^ viable cells were gated for analysis.

Cells stimulation was performed with 5 × 10^5^ PBMCs/well in 96-well round-bottom plates or 1 × 10^5^ Percoll-enriched monocytes/well in 96-well flat-bottom plates. After 24 h of preincubation with Lenvatinib/DMSO, the cells were stimulated with 1 × 10^6^/mL of heat-killed Candida albicans for 7 days or 10 ng/mL of LPS for 24 h. Supernatant was collected and stored at −20 °C until measurement.

Three TC cell lines were used for co-culture assay; these were characterized by different pathological types and mutation status, namely, TPC-1 (RET/PTC rearrangement), BCPAP (BRAF V600E mutation), and FTC-133 (PTEN deficient). All cancer cell lines were grown in RPMI 1640 Dutch modification culture medium supplemented with 10% Fetal Bovine Serum. ThinCertTM cell culture inserts were placed in the 24-well plate to create the co-culture setting (Greiner Bio-One GmbH, Kremsmünster, Austria). Cell counts were determined with a Coulter particle counter (Beckman Coulter Inc., Brea, CA, USA). A total of 50,000 TPC-1 cells (or the other TC cell lines, as indicated) were added to the trans-well inserts (the upper chamber). The same number of monocytes were added into trans-well inserts as the control. After overnight attachment, TC cell lines were co-incubated with 5 × 10^5^ monocytes in the lower chamber in 500 µL of medium containing 1 µM of Lenvatinib or DMSO vehicle control. In some experiments, tumor cells were pre-treated with Lenvatinib in serum-free medium for 6 h. Prior to co-culture, medium containing Lenvatinib was removed, cells were washed with PBS, and fresh medium without Lenvatinib was added into the upper and lower chambers; this was termed as the pre-treatment group. After 24 h of co-culture, the trans-well inserts were discarded, and monocytes were stimulated as previously described.

GM-CSF/M-CSF differentiated macrophages were generated by culturing Percoll-enriched monocytes in 10% HPS RPMI medium containing GM-CSF (10 ng/mL, Miltenyi) or M-CSF (50 ng/mL, Miltenyi) for 6 days. Cells were seeded with 2.5 × 10^6^ per well in a 6-well plate for RNA, Western blot, and seahorse experiments; 5 × 10^5^ cells per well in a 24-well plate for flowcytometry analysis. The cells were detached by gentle scraping and the concentration was adjusted based on the results of flowcytometry (Beckman Coulter Inc.).

### 2.4. Flow Cytometry Analysis

Adherent monocytes/macrophages were harvested by gentle scraping, followed by viability staining. The cells were preincubated at 4 °C for 10 min with staining buffer containing 2% HPS prior to mAb labelling to CD14, CD163, CD206, CD64, CD86, HLA-DR. Intracellular TNF-α was measured in macrophages stimulated for 6 h with LPS (10 ng/mL) in the presence of Golgi plug (BD Bioscience, Franklin Lakes, CA, USA). The cells were fixed and permeabilized (Invitrogen, Carlsbad, CA, USA) followed by incubation with mAb to TNF-α. For the T cell proliferation assay, PBMCs were loaded with CellTrace violet (Invitrogen, Carlsbad, CA, USA) per the manufacturer’s instructions and cultured for 7 days. The CellTrace dilution signal was analyzed on gated CD3^+^ cells. Data were acquired using Cytoflex (Beckman Coulter Inc.) and analyzed with FlowJo V10 software (Tree Star, Inc., Ashland, OR, USA). The results were presented as geometric mean fluorescence intensity (gMFI) or as the percentage of positive cell populations.

### 2.5. Cytokine Measurements

Cytokine (TNF-α, IL-6, IL-10, IL-17, IFN-γ, IL-22) concentrations in supernatant were determined using a commercial ELISA kit (R&D System, Minneapolis, MN, USA), following the instructions of the manufacturer. 

### 2.6. Metabolic Measurements

GM-CSF differentiated macrophages were cultured as described above and detached on Day 6. A total of 50,000 cells/well were seeded onto overnight-calibrated cartridges in assay medium (DMEM supplemented with 1 mM of pyruvate, 2 mM of L-Glutamine, and 1 mM of glucose for the Mito Stress Test; DMEM supplemented with 1 mM of L-Glutamine for the Glyco Stress Test, pH = 7.4) and incubated for 1 h in a non-CO_2_ corrected incubator at 37 °C. The oxygen consumption rate (OCR) and extracellular acidification rate (ECAR) were analyzed using the Mito Stress and Glyco Stress Test kit in an XFp analyzer (Seahorse bioscience, Copenhagen, Denmark). The Mito Stress Test was performed with sequential injections of mitochondrial respiration inhibitors at final concentrations of 1 μM of oligomycin, 1 μM of FCCP, 0.5 μM of Rotenone/Antimycin A. The Glyco Stress Test was performed with 2-DG (50 mM) in the presence of glucose (10 mM). Each condition was tested in 5 replicates. Lactate concentration was measured in serum-free medium using Amplex Red Lactate oxidase assay (Sigma-Aldrich).

### 2.7. RT-qPCR

Total RNA was extracted with the RNeasy Mini kit (Qiagen, Hilden, Germany) and reverse-transcribed into cDNA with the iScript™ kit (Bio-Rad, Hercules, CA, USA). A real-time PCR was performed with the SYBR Green assay (Applied Biosystems, Foster City, CA, USA). Relative quantification was performed using the 2^−ΔΔCt^ method, and RPL37A was used as the housekeeping gene.

### 2.8. Western Blotting

Monocytes or GM-CSF differentiated macrophages were cultured in 6-well plates and lysed with RIPA buffer for Western blot. Protein concentration was determined by a BCA assay. An amount of 20 μg of proteins was loaded onto 10-well pre-casted gradient gels (4–15%) (Bio-Rad). Protein was transferred using the Trans-Blot Turbo Transfer system (Bio-Rad), after which PVDF membrane was incubated overnight at 4 °C with monoclonal antibodies against VEGFR1 and FGFR1. β-actin were used to normalize the protein expression.

### 2.9. Kinase Inhibition Experiments

Kinase inhibitors SB 202190(p38 MAPK inhibitor), U0126 (ERK1/2 inhibitor), SP 600125 (JNK inhibitor), and R406 (Syk inhibitor) (Sigma-Aldrich) were dissolved in sterile DMSO and stored at −20 °C. Monocytes were treated with kinase inhibitors (0, 1 µM, 10 µM) with or without Lenvatinib (1 µM) for 4 h or 24 h before LPS stimulation. Cytokine production was determined as previously described.

### 2.10. Statistical Analysis

The statistical analysis was performed with Prism 9 (GraphPad Software Inc., La Jolla, CA, USA). Nonparametric Wilcoxon matched-pair tests were used to compare means. Data are shown as means ± SEM. * *p* < 0.05, ** *p* < 0.01, *** *p* < 0.001.

## 3. Results

### 3.1. Lenvatinib Modulates the Function of Monocytes

As a first step to evaluate whether Lenvatinib interferes with immune cell function, we performed a dose–response test on human circulating immune cells in vitro. The physiologically relevant dose range was determined based on pharmacokinetic data from Phase I/II clinical trials. Plasma maximum and trough concentrations were reported as 598.0 ng/mL (1401 nM) and 33.9ng/mL (79.4 nM), respectively [25,26]; hence, 100 nM to 1 µM was deemed as a clinically meaningful dose range in our study The immune response was determined by measuring the ROS production from neutrophils, and the secretion of monocyte-derived (TNF-α, IL-6) and lymphocyte-derived (IL-17, INF-γ, IL-22) cytokines upon stimulation (Figure 1A).

No significant changes in neutrophil ROS production were observed, either by pathogen-specific (zymosan) or non-specific (PMA) stimulation (Figure 1B). Moreover, no spontaneous cytokine secretion was detected in monocytes or PBMCs incubated with Lenvatinib for 24 h or 7 days. Interestingly, the production of TNF-a and IL-6 elicited by LPS was largely attenuated by Lenvatinib in a dose-dependent manner (Figure 1C). In contrast, no detectable effect on Lenvatinib on either T cell proliferation (Appendix A) or IL-17 and IFN-γ production was observed after a 7-day Candida stimulation, except a slight increase in IL-22 at the highest Lenvatinib concentration (Figure 1D). Taken together, these data suggest that monocytes are particularly susceptible to Lenvatinib treatment, whereas lymphocytes and neutrophils are less affected.

### 3.2. Lenvatinib Partially Reverses the TC-Induced Monocyte Phenotype

Noting that Lenvatinib affects monocytes stimulated with exogenous ligands, we further investigated whether the functional rewiring of monocytes in the context of TC is also affected by Lenvatinib treatment. Therapeutic perturbation of the TC secretome by Lenvatinib might interfere with the functional and phenotypic differentiation of TAMs [27]. To test this hypothesis, Lenvatinib was added into the trans-well co-culture system, which aims to mimic the cell-to-cell communication within TME.

In these sets of experiments, we observed the spontaneous production of IL-6 and IL-8 from the TPC-1 cell line, whereas TNF-α and IL-10 were below the detection limit (data not shown). To exclude the possibility of any influence of tumor-derived cytokines on the interpretation of monocytes’ immune response, TNF-a and IL-10 were used as the major readout of cytokine response. In agreement with previously published data [28], after 24 h of co-culture with TPC-1 cells, monocytes showed a higher expression of immunosuppressive surface markers, CD163 and CD206. Concomitantly, a pro-inflammatory cytokine response was observed, which was demonstrated as an elevated TNF-α and downregulated IL-10 level (Figure 2B). This points towards a mixed M1/M2 phenotype which resembles TAMs [29]. Having validated the efficacy of the co-culture model, we applied this method to evaluate the role of Lenvatinib in regulating TAMs induction. Interestingly, Lenvatinib significantly attenuated TNF-a production, while it increased IL-10 release. Furthermore, the expression of the M2 surface marker, CD206, was diminished by Lenvatinib, which was shown as a lowered percentage of CD206^+^ CD14^+^ cells. CD163 is generally expressed on almost all the CD14^+^ monocytes at baseline; however, the signal intensity was attenuated by Lenvatinib (Figure 2D). In the meantime, the M1 markers (CD64, CD86, and HLA-DR) remained unchanged (Figure 2E). Flowcytometric cell discrimination and viability measurements revealed that cell viability was not affected after 24 h exposure to Lenvatinib within the clinically relevant dose range (100 nM to 1 µM) (Figure 2C).

The fact that Lenvatinib concomitantly affected monocyte function raised the question regarding to what extent the alteration in TPC-1 secretome could explain the immunological effects we observed. To address this, we tested two additional TC cell lines, FTC-133 and BCPAP, in the co-culture setting. Previous reports have shown that the growth rate of TPC-1, which bears a Lenvatinib-targeted RTK mutation-RET rearrangement, is strongly inhibited by Lenvatinib, whereas FTC-133 and BCPAP, negative for Lenvatinib-targeted mutations, are not sensitive to the drug treatment [30]. Strikingly, similar changes in cytokine production caused by Lenvatinib were observed across all three co-cultures (Figure 3A). On the other hand, a TC-pre-treatment group was included as the control, where only TPC-1 cells had been pre-exposed to Lenvatinib for 6 h before starting the co-culture. Interestingly, incubation with Lenvatinib-pre-treated TC cells resulted in a similar (albeit less strong) effect on cytokine production, suggesting that Lenvatinib partially abolished the pro-inflammatory and tumor-promoting phenotype induced by TPC-1 (Figure 3B). These data demonstrated that the tumor cell-mediated mechanism alone was insufficient to explain the immunomodulatory nature of Lenvatinib in the co-culture experiments, arguing for the direct immunological effects of Lenvatinib on immune cells. These results were further supported by the similar results from GM-CSF- or M-CSF-primed monocytes in the absence of tumor cells (Figure 3C), which is suggestive of a more critical role of the monocyte-intrinsic mechanism underlying this effect.

### 3.3. Prolonged Exposure to Lenvatinib Affects Macrophage Survival and Differentiation

We next investigated whether Lenvatinib could affect the survival and phenotypic differentiation of macrophages. To address this, monocytes were cultured with either GM-CSF (M1 differentiation) or M-CSF (M2 differentiation) for 6 days in the presence of Lenvatinib or a DMSO control. We found that long-term exposure to Lenvatinib impaired macrophage survival, with M-CSF-induced macrophages being more vulnerable to Lenvatinib, resulting in approximately half of the cells being lost during the differentiation process (Figure 4A). More importantly, we observed drastic morphological changes in Lenvatinib-treated macrophages. As indicated in the flowcytometry FSC channel, the cells were significantly smaller (Figure 4B), suggesting the potential impairment of macrophage differentiation. 

Cytokine production was determined after the adjustment of cell concentration, despite a prototypical anti- and pro-inflammatory cytokine production pattern upon LPS stimulation from GM-CSF/M-CSF primed macrophages. The modulation of TNF-a and IL-10 production induced by Lenvatinib was similar in both groups (Figure 4C). This was further supported by intracellular TNF-α level measurement (Figure 4B), recapitulating our observations from monocytes. 

Metabolic reprogramming shapes a distinct immune response to stimulus [31]. To explore the cellular metabolic mechanisms that might contribute to the cytokine production pattern, we quantified the indicators of mitochondrial respiration and glycolytic capacity by oxygen consumption rate (OCR) and extracellular acidification rate (ECAR) using seahorse. Lenvatinib-treated macrophages displayed a trend towards suppressed metabolic activities, which is shown as lower basal and maximal respiration, lower glycolysis activity and glycolytic capacity, while the glycolytic reserve remains unchanged (Figure 4E). Consistent with these findings, the Lenvatinib-treated group produced lower amounts of lactate upon 24 h of LPS challenge (Figure 4D), which was indicative of general defects in both oxidative phosphorylation and glycolytic activities [32].

Taken together, these data demonstrate that Lenvatinib exerts multifaceted inhibitory effects on macrophage survival and maturation.

### 3.4. The Immunological Effects on Monocytes Are Independent of the Lenvatinib-Targeted Receptor Tyrosine Kinase (RTK) Pathways

To explore the underlying mechanism of Lenvatinib-mediated immunological effects, we evaluated the expression of Lenvatinib-targeted RTKs on myeloid cells. Mining into the publicly available transcriptomic database Human Protein Atlas [33], we found that most of the Lenvatinib targets were not detected in circulating myeloid cells, except VEGFR1 and FGFR1, with a relatively low level of mRNA expression in classical monocytes, shown as normalized transcript per million (nTPM) (Figure 5A). To further validate the expression of VEGFR1 and FGFR1, we performed RT-qPCR and Western blotting using samples harvested from buffy coat monocytes and 6-day differentiated macrophages. Despite comparatively consistent mRNA expression in both monocytes and macrophages (Figure 5B), the Western blots did not show any detectable VEGFR1 and FGFR1 expression on monocytes (Figure 5C). Therefore, it is hard to explain the Lenvatinib-mediated mediated immunomodulation by its on-target activity, pointing instead to potential off-target mechanisms. 

To elucidate the potential alternative mechanisms of the action of Lenvatinib, we explored the ProteomeXchange database (dataset PXD005336), a public resource containing the target landscape of 243 clinically tested kinase drugs. The data were obtained via systemic kinase assays using four cancer cell lines [34]. Surprisingly, we found that Lenvatinib interacts with multiple unanticipated kinases (Appendix A). Among all the additional targets, several mitogen-activated protein kinases (MAPKs) were of particular interest. The sequential induction of pro- and anti-inflammatory cytokine by LPS requires fine-tuned control via MAPKs, consisting of extracellular signal-regulated kinases1/2 (ERK1/2), JUN N-terminal kinase (JNK), and p38 MAPKs [35,36,37]. In addition to the MAPK family, the tyrosine kinase Syk regulates LPS-mediated TLR4 endocytosis [38]. If the TLR4 downstream kinases are responsible for Lenvatinib-induced immunomodulation, then the blockage of a certain kinase pathway with highly selective inhibitors could potentially abolish Lenvatinib’s influence on monocytes. To test this hypothesis, we combined Lenvatinib with different inhibitors against p38 MAPK, ERK1/2, JNK, and Syk, respectively [39]. As expected, the inhibition of p38 MAPK, ERK1/2, and Syk alone was able to attenuate TNF-α release. More interestingly, the reduction in TNF-α production caused by Lenvatinib was completely abrogated by concomitant p38 MAPK inhibition, whereas Lenvatinib’s effect persisted in the presence of the ERK1/2, JNK, and Syk inhibitors (Figure 6A–D). Despite this, p38 MAPK blockage alone was not able to recapitulate the IL-10 response induced by Lenvatinib. Strikingly, the Syk inhibitor, R406, resulted in enhanced IL-10 production by itself, and the dual treatment of R406 plus Lenvatinib did not further increase the IL-10 level [40]. Furthermore, we observed similar effects in GM-CSF/M-CSF primed monocytes ( Appendix A). Therefore, Lenvatinib’s ability to block p38 MAPK and Syk appears to be the possible mechanism underlying its opposing effect on TNF-α and IL-10 production.

## 4. Discussion

In the present study, we show that circulating monocytes are particularly vulnerable to ex vivo Lenvatinib treatment at the concentrations equivalent to pharmacological concentrations achievable during treatment, with the function of neutrophils and lymphocytes being less affected. Specifically, the induction of pro-inflammatory monocytic cytokine response was largely attenuated in the presence of Lenvatinib. Furthermore, the drug also interferes with the proinflammatory potential of TC-induced TAMs. Critically, these effects could not be explained by the drug impact on the TPC-1 tumor cells, but rather through direct effects on immune cells. The direct drug impact on monocytes is further supported by in vitro experiments of macrophage differentiation, showing effects of Lenvatinib on cell survival, morphology, cytokine response, and metabolism. Strikingly, we observed no detectable expression in the monocytes of Lenvatinib’s classical targets. On the other hand, blockage of the p38 MAPK and Syk pathways partially diminished Lenvatinib’s immunological effects. Thus, we propose that Lenvatinib directly impacts the functional status of human monocytes via non-canonical mechanisms.

Due to the importance of tumor–immune cell interaction, we initially hypothesized that Lenvatinib might interfere with TAMs-induction by an indirect tumor-instructed mechanism. Previous reports using murine models have shown that TKIs prompt tumor cells to release chemotactic factors (CXCL12 and HMGB-1) or induce immunogenic cell death, resulting in robust immune cell infiltration and tumor clearance [16,41]. Contrary to this, in our co-culture experiments, we did not observe significant differences between the settings with or without the presence of tumor cells. One of the possible explanations is that the tumor-killing capacity of Lenvatinib in vivo relies more on its anti-angiogenetic effects, which require the presence of TME. Supporting this, clinical trials revealed that tumor response to Lenvatinib was not correlated with targeted RTK mutation status [7,42]. Therefore, we were unable to fully recapitulate the tumor response to Lenvatinib with simple two-factor co-culture experiments. Future in vivo studies will be required to elucidate the possible tumor-mediated mechanism.

The direct influence of Lenvatinib on monocytes was interesting and unexpected. Recent advances in drug–target interaction led to the realization that the majority of previously established TKIs bind to additional kinases, including serine-threonine kinases that are not closely related to the function and sequence of the intended targets [34,43]. However, their immunological implications remain largely unexplored, and preclinical results reflect that the immunologic effects of individual TKIs are largely drug-specific and context-dependent [19,44,45].

Lenvatinib is a multitargeted kinase inhibitor with a particularly potent activity against the VEGFR2 via type V binding model [46]. The side effects of Lenvatinib, such as hypertension, are predominantly caused by its anti-angiogenetic activity [4], while no major immune-related side effects have been reported. It might be reasonable to postulate that Lenvatinib-mediated immunological effects are well tolerated by patients, yet they have been largely overlooked since the early stage of drug development [2]. No data on intratumoral Lenvatinib concentrations have been reported; thus, the working concentration in this study was based on plasma pharmacokinetic data. We selected 1 µM as the highest non-cytotoxic concentration and observed that monocyte activation status was significantly influenced at this concentration. In line with this, a recent study also demonstrated that, compared to other immune cells, monocytes were less resistant to in vitro TKIs treatment [47]. Similarly, a clinical observational study also reported decreased levels of circulating TNF-α and IL-6 in hepatocellular carcinoma (HCC) patients after Lenvatinib treatment [48].

As for the mode of action, it appears plausible that Lenvatinib might exert inhibitory activity through p38 MPAK and Syk kinase in monocytes. We found that the inhibition of p38 by SB202190 completely abolished the effect of Lenvatinib on TNF-α production. In line with our findings, it has been previously described that, in tumor cells, Lenvatinib directly binds with several components within the MAPK signaling pathway, including MAP4K2, MAP4K5, MAPK14, and MAPKAPK2 [49,50]. p38 MAPK plays a pivotal role in governing macrophage alternative activation and preventing apoptosis [51,52]. Indeed, we observed the selective downregulation of M2 cell surface markers on TC-induced monocytes and significant cell loss caused by Lenvatinib during macrophage differentiation. Additionally, a recent study investigating the immunological effects of Regorafenib, another TKI used for HCC, revealed that its synergizing efficacy with anti-PD1 therapy was associated with the suppression of p38 kinase phosphorylation, which reversed the M2-like polarization of murine bone marrow-derived macrophages (BMDMs) [45], thus supporting the off-target effects via p38MAPK blockage seen in our study.

A striking aspect of the Lenvatinib-treated monocyte is the persistent elevation of IL-10 production across all the experimental settings. The high production of IL-10 and TGF-β is a hallmark of TAMs [53]. However, seemingly counterintuitively, we did not observe the concomitant upregulation of other immunosuppressive markers in Lenvatinib-treated TC-induced monocytes, such as CD163 and CD206. The elevation of IL-10 seemed to be independent of the general immunosuppressive traits. As part of the host defense mechanism, the induction of IL-10 occurs simultaneously with proinflammatory cytokines via shared p38 MAPK and ERK pathways [54]. The opposing effects on IL-10 and TNF-a production indicate additional pathways influenced by Lenvatinib treatment.

Syk is a non-receptor tyrosine kinase transmitting signals of immunoreceptors, regulating the activation of innate immunity [55,56]. We found that Syk inhibitor, R406, was the only agent that could enhance IL-10 production upon LPS stimulation, and no synergizing effect with Lenvatinib was observed. It is tempting to speculate that the Syk pathway is also involved in Lenvatinib’s off-target effects. Consistent with our findings, others have shown that R406 selectively blocks the induction of IL-6 and TNF-a, while restoring the IL-10 production in human monocytes [40]. However, this is at odds with previous reports showing that Syk functions as a negative-feedback mechanism to limit inflammation by promoting IL-10 production [56]. Nevertheless, the Syk inhibitor we used, R406, was proposed to be potent and selective against Syk, but it also binds to several other kinases, including Salt-inducible kinase 2 (SIK2) [34]. SIK2 is one of the most common additional targets of TKIs, as revealed by large-scale drug screening [34]. Due to its restriction on the formation of regulatory macrophages, the inhibition of SIK2 stimulates the production of IL-10 [57]. Importantly, drugs with an SIK2 affinity of greater than 500nM are capable of suppressing TNF-a and upregulating IL-10 in BMDM, recapitulating the cytokine response to Lenvatinib [34]. However, due to the complex crosstalk among signaling pathways and the model we used, it is not possible to make a definitive statement about the off-target effect of Lenvatinib on the p38 MAPK, Syk, or SIK2 pathways. Future work will be required to define the comprehensive drug-kinase activity in immune cells using high-throughput profiling techniques [58].

Clinical evidence highlights the beneficial role of Lenvatinib in combination with immunotherapy in metastatic renal cell carcinoma and endometrial cancer, mirrored by an increased objective response rate and prolonged progression-free survival [59,60]. Recently, clinical trials investigating the safety and efficacy of Lenvatinib plus Pembrolizumab (anti-PD-1 antibody) in advanced DTC (NCT02973997) and HCC (NCT03713593) have also been initiated. However, combination regimens generally lead to more severe side-effects than monotherapies. Therefore, attempts to incorporate TKIs into immunotherapy have been largely limited by an increased incidence of high-grade toxicities [61,62]. Thus, a deeper understanding of the neglected immune-cell intrinsic off-target effect is of great clinical significance. Our study provided preclinical evidence that Lenvatinib has a direct impact on the myeloid compartment of the immune system. Recent studies have demonstrated the deleterious role of inflammation in cancer survival, with proinflammatory mediators playing an important role in angiogenesis, as well as acting as survival factors for cancer cells [63]. Our data, demonstrating the anti-inflammatory effects of Lenvatinib, argue that this effect is likely to contribute to the therapeutic effect of this class of drugs, extending their potential therapeutic applicability in tumors characterized by a prominent inflammatory profile. Future work should characterize the immune landscape in patients receiving Lenvatinib and identify predictive biomarkers for sensitivity to immunotherapies.

## Figures and Tables

**Figure 1 pharmaceutics-15-00412-f001:**
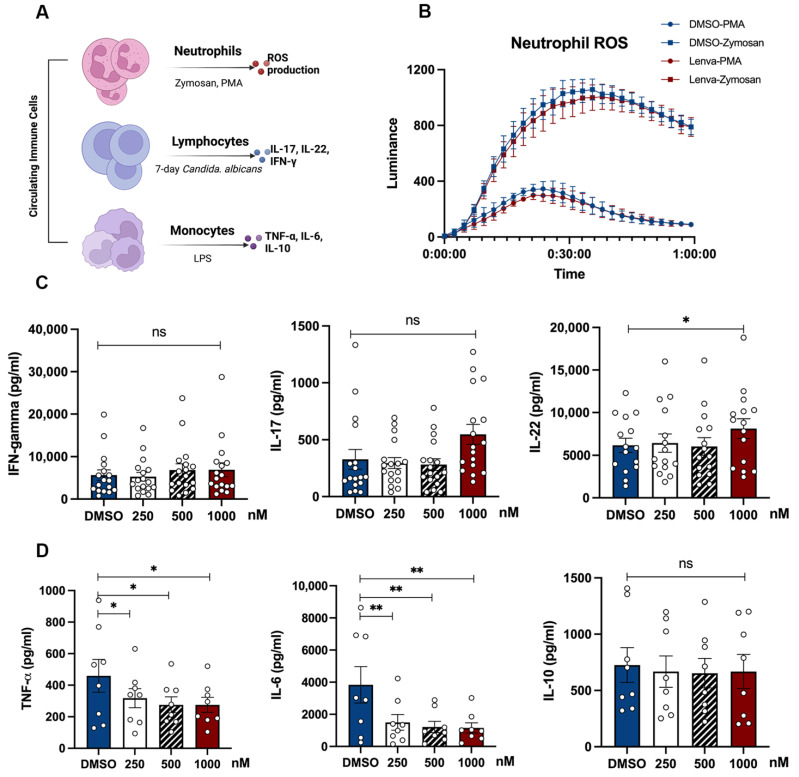
Lenvatinib modulates in vitro monocyte-derived cytokine secretion in dose–response experiments. (**A**) Schematic diagram of the dose–response test. Freshly isolated neutrophils were stimulated with opsonized zymosan or PMA for the measurement of ROS production, after brief exposure to Lenvatinib (1µM) for 1 h. PBMCs from healthy donors were incubated with Lenvatinib with indicated concentrations for 24 h, after which cells were stimulated with LPS or Candida albicans. Cytokines levels in supernatants were determined by ELISA. (**B**) ROS production from neutrophils; data represent mean ± SME at each time point, n = 6. (**C**) Lymphocyte-derived cytokines (IFN-γ, IL-17, and IL-22) after 7-day Candida albicans stimulation, n = 15. (**D**) Monocyte-derived cytokines (IL-6, TNF-α, IL-10) after 24 h of LPS stimulation, n = 8. The bar graphs indicate mean ± SME; each dot represents one healthy donor. * *p* < 0.05, ** *p* < 0.01 by nonparametric Wilcoxon matched-pair tests; ns, not significant; PMA, phorbol myristate acetate; ROS, reactive oxygen species.

**Figure 2 pharmaceutics-15-00412-f002:**
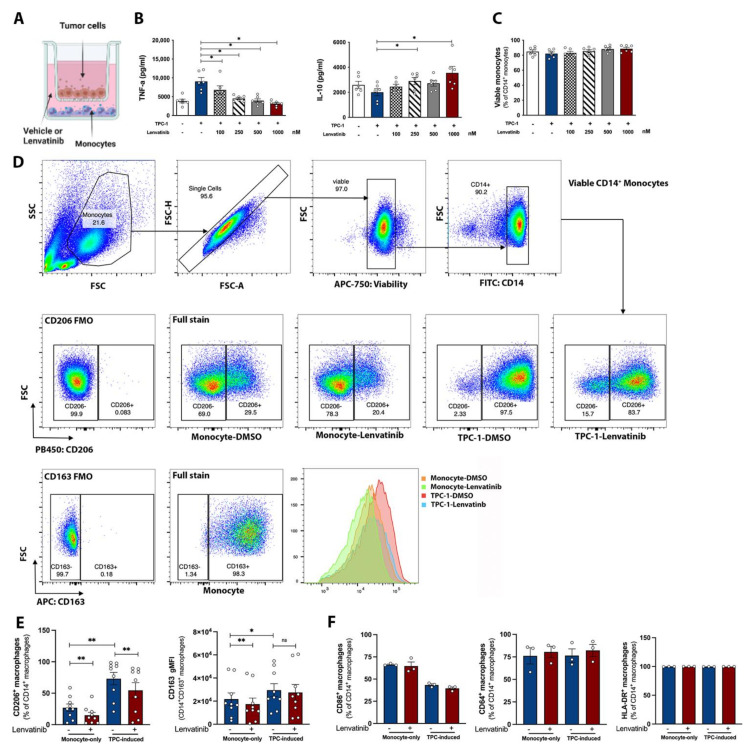
Lenvatinib interferes with monocytes’ functional rewiring in co-culture experiments. (**A**) Schematic diagram of the co-culture experiments. (**B**) Cytokine secretion from TPC-1 induced monocytes treated with indicated concentrations of Lenvatinib; monocytes cultured in RPMI were included as the baseline control, n = 6. (**C**) Viability of monocytes after 24 h exposure to Lenvatinib; percentage of viable CD14^+^ cells were gated for statistical analysis, n = 6. (**D**) Expression of M2 markers in TPC-1-induced monocytes. The upper panel shows the gating strategy to identify the viable CD14^+^ monocytes. Identification of CD206^+^ and CD163^+^ subsets within viable CD14^+^ monocytes was based on FMO controls. The lower panel shows the representative dot plot and histogram from one donor. (**E**) Quantification of the percentage of CD206^+^ subsets within CD14^+^ monocytes and fluorescence intensity of CD163 signal on CD14^+^CD163^+^ monocytes, n = 6. (**F**) Expression of M1 markers (CD86, CD64, and HLA-DR), n = 3. The bar graphs indicate mean ± SME; each dot represents one healthy donor. * *p* < 0.05, ** *p* < 0.01 by nonparametric Wilcoxon matched-pair tests. FMO, fluorescence minus one control; gMFI, geometric mean fluorescence intensity.

**Figure 3 pharmaceutics-15-00412-f003:**
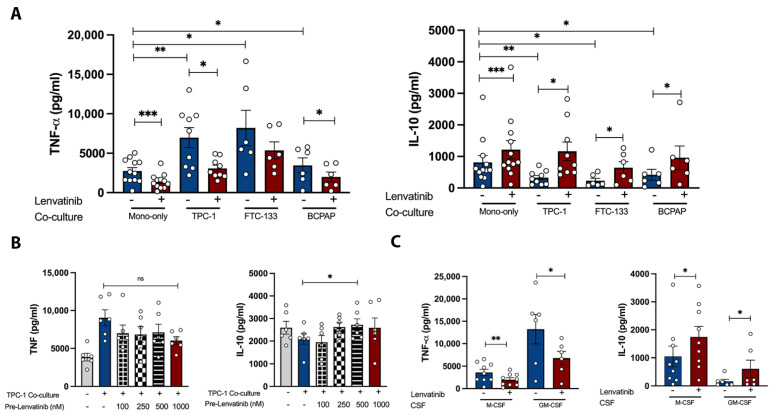
Similar patterns of cytokine production caused by Lenvatinib in different experimental settings. (**A**) In vitro generation of tumor-associated macrophages with 3 TC cell lines. (**B**) Co-culture experiments with TPC-1 cell line pretreated by Lenvatinib for 6 h. (**C**) Monocytes primed by GM-CSF and M-CSF for 24 h, n = 6. The bar graphs indicate mean ± SME; each dot represents one healthy donor. * *p* < 0.05, ** *p* < 0.01, *** *p* < 0.001 by nonparametric Wilcoxon matched-pair tests.

**Figure 4 pharmaceutics-15-00412-f004:**
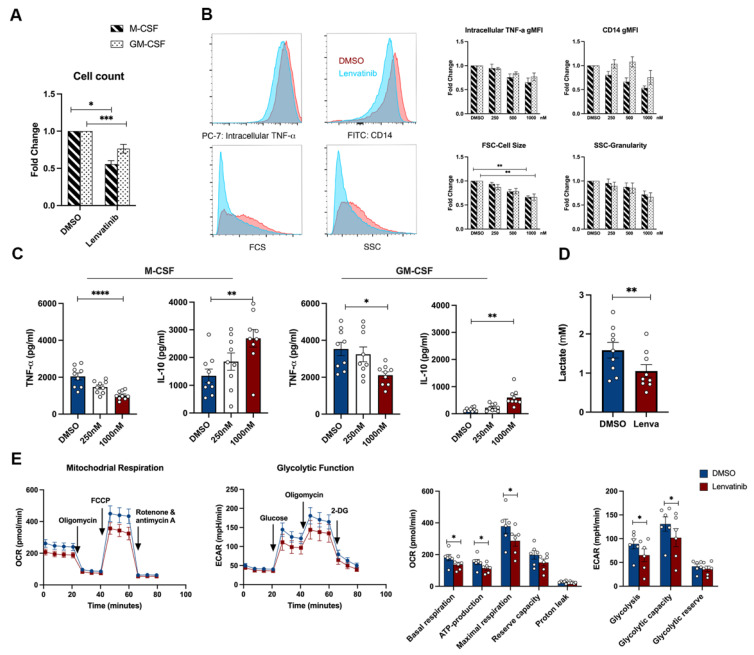
Morphological, functional, and metabolic changes in Lenvatinib-treated macrophages. (**A**) Overall macrophage survival determined by flowcytometry; the cell counts of macrophages were gated on FCS-A and SSC-A (n = 11). (**B**) Left panel indicates representative histogram of the parameters, including intracellular TNF, CD14 expression, cell size (FSC), and granularity (SSC), analyzed by flowcytometry from one donor. The right panel shows the corresponding quantification of gMFI, n = 6. (**C**) Cytokine production from GM-CSF/M-CSF primed macrophages, n = 9. (**D**) Lactate production from GM-CSF-induced macrophages after 24 h of stimulation with LPS, n = 6. (**E**) Analysis of ECAR and OCR from GM-CSF-polarized macrophages was performed with a seahorse XF analyzer. The left figure shows the OCR and ECAR profiles for macrophages exposed to Lenvatinib or DMSO vehicle control. The right figure shows mitochondrial bioenergetic and glycolytic parameters calculated from OCR and ECAR (n = 6). The bar graphs indicate mean ± SME; each dot represents one healthy donor. * *p* < 0.05, ** *p* < 0.01, *** *p* < 0.001, **** *p* < 0.0001 by nonparametric Wilcoxon matched-pair tests. FSC, forward scatter; ECAR, extracellular acidification rate; OCR, oxygen consumption rate; SSC, side scatter.

**Figure 5 pharmaceutics-15-00412-f005:**
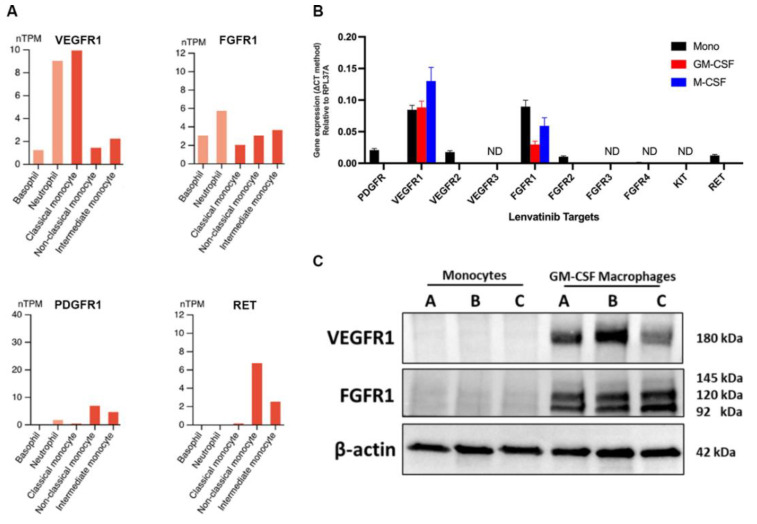
The expression of Lenvatinib-targeted RTKs on monocytes. (**A**) Normalized transcript expression levels of 4 detectable Lenvatinib-targeted RTKs (VEGFR1, FGFR1, PDGFR, RET) from the transcriptome signatures are visualized for different peripheral myeloid cell types from Monaco et al. The panels are screenshots from the Human Protein Atlas resource. (**B**) Quantified Lenvatinib-targeted RTKs mRNA expression by RT-qPCR. Transcript copy number was normalized to cellular control (RPL37A). n = 6. (**C**) Levels of VEGFR1 and FGFR1 in the cell lysate from buffy coat monocytes and GM-CSF polarized macrophages were analyzed by Western blot, and β-actin was used as the loading control. Three isoforms of FGFR1 were detected with molecular weights of 92, 120, and 145 kDa. n = 3. ND, not determined; nTPM, normalized transcripts per million.

**Figure 6 pharmaceutics-15-00412-f006:**
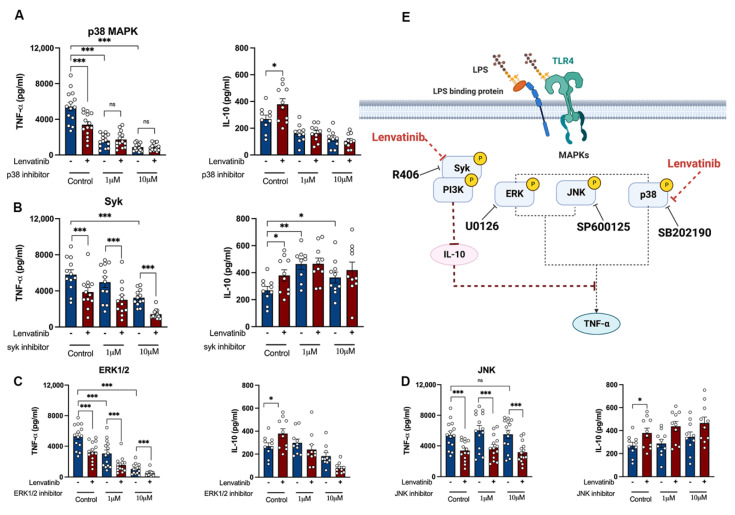
Effects of TLR4 downstream kinase inhibitors on Lenvatinib-mediated immunomodulation. (**A–D**) Monocytes were treated with selective inhibitors against kinases within the TLR4 signaling networks: SB 201019 as p38 MAPK inhibitor, R406 as Syk inhibitor, U0126 as ERK1/2 inhibitor, SP 600125 as JNK inhibitor. All the inhibitors were tested at two concentrations (1 µM and 10 µM), either used alone or in combination with Lenvatinib. Cytokine response was determined by ELISA after 24 h LPS stimulation. Kinase inhibitor single agent versus DMSO control was compared, and kinase inhibitor plus Lenvatinib versus kinase inhibitor alone was compared, n = 10 to 14. (**E**) Schematic diagram of the possible mechanism behind Lenvatinib’s off-target effect on monocytes via dual inhibition of p38 and Syk pathways. The bar graphs indicate mean ± SME; each dot represents one healthy donor. * *p* < 0.05, ** *p* < 0.01, *** *p* < 0.001 by nonparametric Wilcoxon matched-pair tests. ERK, extracellular signal-regulated kinases; JNK, c-Jun N-terminal kinases; ns, not significant; p38 MAPK, p38 mitogen-activated protein kinases; Syk, spleen tyrosine kinase; TLR4, toll-like receptor 4.

## Data Availability

The data presented in this study are available on request from the corresponding author. The data are not publicly available due to privacy issues.

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
