# Peer review of "Influence of Lenvatinib on the Functional Reprogramming of Peripheral Myeloid Cells in the Context of Non-Medullary Thyroid Carcinoma"

_pharmaceutics, 2023, doi:10.3390/pharmaceutics15020412_

Round 1

Reviewer 1 Report

In this study, Peng and colleagues have indicated that Lenvatinib-mediated immunomodulation in monocyte/macrophage. This study will provide the important knowledge on Lenvatinib-induced immunological effects in thyroid carcinoma through p38 MAPK and Syk pathways. However, following comments should be further seriously considered.

1.     In Fig. 1, the author examined the ROS production in neutrophil and cytokine production in monocyte after treatment with PMA and LPS, respectively. But it is generally accepted that LPS can induce ROS production and cytokine production in neutrophil. If authors conclude that the immunological effect of Lenvatinib is specific function in monocyte/macrophage, they should examine whether Lenvatinib affects LPS-induced cytokine or ROS production in neutrophil. This is very important in this study. Is this observation specific to monocyte among immune cell as you mentioned?

2.     The author examined the effect of p38 inhibitor and Syk inhibitor on LPS-induced cytokine production in monocyte. Could p38 inhibitor and Syk inhibitor affect the cytokine production or the population of M2 macrophage after treatment with Lenvatinib in GM-CSF primed monocyte or tumor associated macrophage?

Reviewer 2 Report

I positively appreciate the idea of ​​this research and the way it was carried out, as well as the clear and logical writing of the article. The consistent results are interesting and eloquent and are well integrated in the current bibliography. An important key element of the discussion part of the manuscript is the support of mechanism of the off-target immunological effects of Lenvatinib observed in this research, through the double effect on p38 MAPK and Syk pathways. The authors argue that these results could represent an important step in strengthening the future directions of new treatment strategies involving the combination of Lenvatinib with other immunotherapeutic agents. I have no requests for revision of the manuscript, and I think it can be published in this form.
